

# Observations of PAN and its confinement in the Asian Summer Monsoon Anticyclone in high spatial resolution

J. Ungermann[1], M. Ern[1], M. Kaufmann[1], R. Müller[1], R. Spang[1], F. Ploeger[1], B. Vogel[1], and M. Riese[1]

[1]Institut für Energie- und Klimaforschung – Stratosphäre (IEK-7), Forschungszentrum Jülich GmbH, Jülich, Germany

*Correspondence to:* J. Ungermann (j.ungermann@fz-juelich.de)

**Abstract.** This paper presents a set of observations by the CRISTA infrared limb sounder in low-earth orbit taken in August 1997 and analyses of trace-gases in the Asian Summer Monsoon (ASM) region. The spatially highly-resolved measurements of peroxyacetyl nitrate (PAN) and $O_3$ allow a detailed analysis of an eddy-shedding event of the ASM anticyclone. We identify enhanced PAN volume mixing ratios (VMRs) within the main anticyclone and within the eddy, which are suitable as a tracer

for polluted air within the ASM originating in India and China. Comparing the retrieved PAN VMRs with potential vorticity (PV) on isentropes reveals that the PAN VMRs exhibit the strongest decrease at each isentrope for an increasing value of PV which may be used to identify the extent of the ASM. Using temperature values also derived from CRISTA measurements, we also computed the location of the thermal tropopause according to the WMO criterion and find that its location agrees well with the limits of the area of increased PAN VMRs both horizontally on isentropes and vertically within the anticyclone.

In contrast, the shed eddy exhibits enhanced PAN VMRs for 1 to 2 km above the thermal tropopause. Using the relationship between PAN as a tropospheric tracer and $O_3$ as a stratospheric tracer to identify mixed air parcels, we further found the anticyclone to contain few such parcels, whereas the region in between the anticyclone and the eddy contains many mixed parcels. In combination, this implies that while the anticyclone confines polluted air masses well, eddy shedding provides a very rapid horizontal transport pathway of Asian pollution into the extratropical lowermost stratosphere with a time scale of

only a few days.


# 1 Introduction

The Asian Summer Monsoon (ASM) anticyclone is the dominant circulation pattern in the summertime Northern Hemisphere in the upper troposphere / lower stratosphere (UTLS) and has a major impact on trace gas composition and stratosphere-troposphere exchange (STE). It is caused by persistent thermal heating during summer with a thermal low at lower altitudes and a strong anticyclone in the upper troposphere (UT) (e.g. Krishnamurti and Bhalme, 1976). It can be found in summer in the larger vicinity of Tibetan plateau and is enclosed by the subtropical jet on its northern side and the equatorial easterly jet on its southern side (e.g. Randel and Park, 2006). These jets act as a strong transport-barrier and prevent the air inside the anticyclone from mixing along isentropes into the extratropical lower stratosphere or the equatorial tropics (e.g. Park et al., 2008). Previous studies using satellite data have thus found an enhancement of carbon monoxide (e.g. Li et al., 2005; Park et al., 2007) and other trace gases in the ASM indicating that polluted air is trapped within the anticyclone below the thermal tropopause (Park et al., 2008). The ASM is largely responsible for the moistening of the lowermost stratosphere in summer (Gettelman et al., 2004; Ploeger et al., 2013). It is however not fully clear, how much of the STE occurs isentropically, by eddy-shedding, or vertically across the tropopause (e.g. Hsu and Plumb, 2000; Randel et al., 2010; Vogel et al., 2015; Garny and Randel, 2015).

The CRyogenic Infrared Spectrometers and Telescopes for the Atmosphere (CRISTA; e.g. Offermann et al., 1999) experiment was flown on two Space Shuttle missions (STS66 and STS85) in November 1994 and August 1997, respectively. During free flying periods of about 10 days, global observations of temperature and numerous trace gases were made with unprecedented and so far unmatched spatial resolution (Riese et al., 1997, 1999). It was designed to examine dynamical processes primarily in the stratosphere and mesosphere, but measured also the UTLS region.

By using three viewing directions simultaneously, a horizontal across track sampling of 600 km was achieved. The along track sampling was 200 to 400 km, depending on the measurement mode. The vertical sampling was 1.5 km in the first mission and 2 km in the second mission (Grossmann et al., 2002). The spectral resolution was sufficient to determine 25 trace gas species (Offermann et al., 1999). While more recent satellites offer marginally better vertical resolution in the UTLS (especially solar occultation instruments such as ACE-FTS; Bernath et al., 2005) or a higher spectral resolution (such as MIPAS; Fischer et al., 2008), no other instrument since has offered such a spatial measurement density.

Here, we will examine the confinement of polluted air within the ASM by means of trace gases retrieved from CRISTA-2 measurements (that is measurements taken during the second CRISTA mission) taken in August 1997. We focus on peroxy-acetyl nitrate (PAN) measurements, which offer the highest contrast between tropospheric air within the anticyclone and clean air outside compared to other tracers available from CRISTA measurements.

PAN is a tracer of tropospheric pollution. It is a secondary pollutant that forms in the troposphere from precursors with their main sources being biomass burning and anthropogenic pollution (Stephens, 1969). Its lifetime is highly temperature dependent and it may be only seconds in the boundary layer, but close to the cold-point tropopause, PAN can accumulate for weeks or months (Roberts, 1990). The combination of the relatively long lifetime of PAN at tropopause altitudes together with the confinement of air in the ASM anticyclone is expected to lead to an accumulation of PAN in the ASM circulation. Further sources of PAN are thunderstorms in convective systems that produce $NO_x$ by lightning. The typical sinks are thermal decomposition





at lower altitudes and photolysis in the stratosphere. Due to the low temperatures prevalent in the ASM anticyclone, PAN is well suited to identify polluted air masses that have been uplifted in the current season. For example, PAN was also used in model studies to determine the influence of regional emissions on the air within the ASM (e.g. Fadnavis et al., 2014). The first global dataset of PAN retrieved from satellites was provided by Glatthor et al. (2007) covering the time frame from 2002 until

the end of the MIPAS experiment in 2012. The newly derived data presented here shows the global state 5 years earlier, even if only for one week, and thus extends our knowledge of the historical evolution of PAN in the UTLS considerably.

## 2    Measurements and model data

The measurements presented in this paper were made aboard the Space Shuttle experiment CRISTA. The limb sounder CRISTA was integrated into the astronomical Shuttle Pallet System (ASTRO-SPAS; Wattenbach and Moritz, 1997) and launched into

orbit by the NASA space shuttle for two missions; once in November 1994 (CRISTA-1) and once in August 1997 (CRISTA-2). The instrument employs three Herschel telescopes (pointing towards -18°, 0°, and 18° in reference to the viewing direction, which varied dependent on the latitude between 108° and 252° with respect to flight direction (Grossmann et al., 2002)), four Ebert-Fastie spectrometers (Fastie, 1991) and 24 mid-infrared detectors ranging from 155 to 2390 $\mathrm{cm}^{-1}$ and covers an altitude range from $\approx 9$ to 150 km altitude in the limb. Here, we concentrate for practical matters on the detector channels

covering the spectral range from 777 to 862 $\mathrm{cm}^{-1}$ (with a spectral resolution of $\lambda/\Delta\lambda \approx 500$) and the altitude range between 9 and 45 km. The field-of-view has a size of 3 arcmin×30 arcmin, which corresponds to about 1.7 km×17 km at 18 km tangent height, whereby spectra were usually measured in 2 km altitude steps. As the detectors and the optics were cooled by cryogenic helium, a high measurement speed could be achieved with a simultaneous good signal-to-noise-ratio — recording one spectrum took only 1.2 s allowing the measurement of full profiles in less than one minute with four spectrometers operating in parallel.

The vertical sampling is one spectrum every 2 km, which, taking field-of-view and processing into account, translates to a vertical resolution of the retrieved products of $\approx 2$ km for temperature and $O_3$ and $\approx 3$ km for PAN in the UTLS. As the retrieval grid is aligned with the tangent altitudes of the measured spectra, there is no loss of resolution due to interpolation. No limb-sounder since has achieved a comparable spatial measurement density at this vertical resolution.

Previous work on CRISTA concentrated mainly on the exploration of the middle atmosphere, with only few studies looking

on the UTLS region (e.g. Spang et al., 2015). Here, we examine the Asian summer monsoon (ASM), which could be observed for 8 consecutive days from 8 August 1997 to 16 August 1997 during the CRISTA-2 mission.

The calibrated spectra were available from previous studies (Spang et al., 2015). The level 2 processing was redone based on previous work for the atmospheric successor instrument CRISTA-NF (e.g. Ungermann et al., 2012). The JUelich Rapid Spectral SImulation Code V2 (JURASSIC2; e.g. Hoffmann et al., 2008; Ungermann, 2013) forward model is used to simulate radiances

for given atmospheric states, while the Juelich Tomographic Inversion Library (JUTIL; e.g. Ungermann et al., 2015) performs the actual inversion to derive atmospheric quantities from measured infrared radiances. The setup was slightly modified from the one used for the CRISTA-NF retrievals to accommodate the fact that the CRISTA spectra are interpolated onto a fixed 0.35 wavenumber grid and that the spectral range of the detector was smaller. The resulting setup allows temperature, water



vapour, ozone, nitric acid, peroxyacetyl nitrate (PAN), carbon tetrachloride, CFC-11, CFC-113, and HCFC-22 to be derived, as well as an aerosol underground. Further quantities could be derived by incorporating spectral data acquired by other CRISTA detectors. The retrieval process is briefly described in App. A. An exemplary profile of PAN and diagnostic data averaged over retrieved profiles in the ASM region is shown in Fig. 1.

For analysis of the meteorological setting (pressure, temperature, wind speeds) and also as initial guess (temperature) and a priori (temperature and pressure) for the retrieval, ERA-Interim data supplied by the European Centre for Medium-range Weather Forecasts (ECMWF) were used (Dee et al., 2011). For August 1997, these data are available in six hour time steps in the T255/L60 resolution. Winds from this dataset were also used for calculation of air parcel trajectories with the Chemical Lagrangian Model of the Stratosphere (CLaMS; e.g. McKenna et al., 2002; Konopka et al., 2010).

## 3   Analysis

### 3.1   Synoptic Situation

The majority of the CRISTA-2 measurements in the altitude region of the ASM falls in the period between the 8th and the 13th of August due to the employed measurement mode. Figure 2 shows the approximate extent of the ASM on the $100 \, \mathrm{hPa}$ surface with low potential vorticity (PV) as a marker for air from inside the anticyclone and contour lines of geopotential altitude showing the geostrophic wind and thus the jets forming the horizontal transport barrier. In the beginning of the considered

period, the approximate location of the ASM is located over the Tibetan plateau; subsequently a large eddy is shed eastwards. On 11 August 1997, a clear separation of air masses between the main anticyclone and the smaller eastward propagating anticyclonic eddy can be observed, with a very faint connection in between. This eddy breaks off and continues moving towards the American coast while descending below the shown pressure surface.

### 3.2   Structures in Derived Trace Gases

We focus here on retrieved PAN VMRs, as this trace gas has the best quality of the derived trace gases with respect of characterising the ASM anticyclone. The positive water-vapour anomaly within the anticyclone has a much smaller gradient to stratospheric air at the same altitude resulting in a smaller signal-to-noise ratio; the same is true for the chlorofluorocarbons. Ozone would also work as a negative anomaly highlighting tropospheric air, but it has sources in both the stratosphere and

the polluted troposphere and has thus also a smaller gradient at the examined altitude levels. We revisit ozone at a later point, though (see below).

Figure 3 shows PAN measurements for the five consecutive days offering the best measurement density in the UTLS owing to the configuration of the CRISTA instrument. The measurement gaps over India and the Pacific are caused by high altitude cloud systems that prevent the retrieval of trace gases from measured spectra at the shown altitudes. To remove cloud-affected spectra

from the retrieval, the colour ratio between the radiances averaged between $791.00 \, \mathrm{cm^{-1}}$ to $793.00 \, \mathrm{cm^{-1}}$ and $832.30 \, \mathrm{cm^{-1}}$ to $834.40 \, \mathrm{cm^{-1}}$ is determined and spectra with a ratio of less than 4 are discarded (Spang et al., 2012). Shown is the $380 \, \mathrm{K}$





surface which is typically located rather high in the anticyclone and is well suited to identify the boundaries of the anticyclone due to expected high horizontal gradient in trace gases (Ploeger et al., 2015).

In the extratropical stratosphere, PAN mixing ratios larger than 50 to 80 pptv are not expected, but filaments with higher VMR may be found in the tropical troposphere and in the extratropical mixing layer. However, within the CRISTA-2 measure-
5 ments over Asia, the air masses with highly enhanced VMRs compared to the stratospheric background are solely found within the anticyclone of the ASM and the spun-off eddy.

The PAN VMR measurements in Fig. 3 plotted at their tangent points sketchily show the eddy shedding event (compare to Fig. 2). Especially the weakening 'connection' at $120°$ E is well covered by measurements on each day. On 10th August 1997, the main anticyclone and the forming eddy are still well connected. The connection progressively weakens until 13th
August 1997, when the eddy is finally well separated by air masses with low PAN VMRs. Note further the long drawn PAN filament stretching towards Alaska on the 12th and 13th of August.

As we may well assume that the PAN VMRs are rather stable for a couple of days at the given altitudes and temperatures, one can improve the measurement density by synoptically interpolating the measurements of multiple days to a single point of time using trajectory calculations. For each measured air parcel, trajectories were computed using the CLaMS trajectory model from
15 the time of measurement to 12:00 UTC of each measurement day. Figure 4 shows PV and corresponding synoptically calculated PAN data on 380 K. Comparing Fig. 3 with Fig. 4 shows that the trajectory-based fill-in makes the prevalent structures more apparent. The negative anomalies in PV agree very well with the positive PAN anomalies and the highest PAN VMRs at this altitude clearly occur for PV values of 3 PVU or less. The sharpest PV gradient occurs towards the northern sub-tropics in agreement with a sharp drop of PAN VMRs towards the extratropical lowermost stratosphere to the north. The PV gradient in
the southern part of the anticyclone is less pronounced and the structures in PAN VMRs are correspondingly less distinct. Only further south, clean tropical air streaming in from the Pacific exhibits PAN VMRs below 50 pptv.

The eddy-shedding event can be seen well in both PV and PAN VMRs. Even the fine filamentary structure in between the anticyclone and the spun-off eddy can be distinguished in retrieved PAN VMRs. We thus conclude that PAN is an excellent tracer to identify anticyclone air in the given time frame (see also Fadnavis et al., 2014).
However to reduce potential artefacts introduced by the synoptic calculation, the PAN VMRs for the figures presented from now on are calculated for the mid-point of the measurement period, thus reducing the time over which trajectories are calculated to less than 36 hours on average.

### 3.3 Horizontal and Vertical Confinement

The 380 K isentrope is well suited in the given meteorological situation to describe the confinement of the polluted air masses
of the Asian monsoon. Figure 5 shows two vertical cross-sections through the potential vorticity field, one through the main anticyclone and one through the eddy. For both PV and PAN, the northern boundary is formed by the jet-stream with strong winds of more than $30 \, \text{ms}^{-1}$ following a sharp increase in PV. The southern boundary coincides with the thermal tropopause on the isentrope close to the equatorial jet.




The relationship between PV and PAN VMRs can be further explored by comparing typical PAN VMRs for each value of PV. Figure 6a shows a scatter plot of PAN VMR against PV. There is a general gradient from high PAN VMRs at low potential temperature values towards low PAN VMRs at high potential temperature values as is expected for a tropospheric pollutant. However the area between 360 K and 400 K has an additional horizontal structure that is related to the ASM anticyclone. To

extract further information from the noisy, unstructured data, a smooth 2-D surface on a regular grid is fitted against the data in Fig. 6b. A fine regular grid is defined in potential temperature and modified PV. Modified PV ($\mathrm{PV_{mod}} = \mathrm{PV} \cdot (\theta/380\,\mathrm{K})^{-4.5}$) is a scaled form of PV that partially reduces its dependency on potential temperature (Lait, 1994; Müller and Günther, 2003) and thereby transforms the covered tracer-PV space into a more rectangular region (the choice of the exponent is not very important for this purpose). The values on the grid are then determined by least-squares minimisation against the measured data

with additional smoothness constraints on the first derivative with respect to potential temperature and the second derivative with respect to modified PV. The strength of the constraints somewhat influences the derived absolute values, but the shape and the position of the maxima are robust. Figure 6c shows that the PV value, for which PAN decreases most strongly, increases smoothly and constantly with potential temperature up to around 400 K, where the PAN anomaly slowly vanishes at thermal tropopause level. This plot enables the visualisation of the confinement of the ASM anticyclone and which air masses are

entrapped within. The position of the transport barrier can be easily discerned by the negative anomaly in Fig. 6c. The positive anomaly at lower potential temperatures is caused by a set of parcels of unpolluted air within the tropical jet.

Figure 7 shows that the decrease of PAN VMRs for increasing PV is strongest between 370 K and 375 K with a maximum decrease of $\approx -60\,\mathrm{pptv}\,\mathrm{PVU}^{-1}$. On the 380 K isentrope, the maximal gradient is given by $\approx -44\,\mathrm{pptv}\,\mathrm{PVU}^{-1}$ at $\approx 3.2\,\mathrm{PVU}$. This is consistent with the study of Ploeger et al. (2015), where a similar comparison for modelled CO and PV exhibited peak

gradients between 2.6 PVU and 4.6 PVU for the years of 2011 to 2013. As the southern transport barrier coincides with the thermal tropopause, it is plausible that a value between 2 PVU and 4 PVU is found for the maximum gradient because PV values in this range are also commonly chosen for the dynamical tropopause (e.g. Kunz et al., 2011).

In Fig. 8, we again consider the PAN distribution at the 380 K surface. Comparing the PV value for the maximum gradient (dPAN/dPV) at 3.2 PVU with the given PAN VMRs gives a mixed picture. It matches indeed very well the maximum gradient,

where PAN VMRs fall from values of 200 pptv to below 130 pptv, but both in the anticyclone and in the eddy shedding region, it seems too conservative, resulting in a too small area that would be attributed to the anticyclone. The PAN VMRs may be not well-mixed enough within the anticyclone such that the steepest decrease in VMR coincides with the ASM boundary. This suggests that the PV value for the *maximal* gradient may not be the best PV value for discerning the boundaries of the ASM for less well-mixed trace gases (the criterion was developed for CO, which has a longer life time than PAN). A slightly larger

value such as 3.7 PVU (incidentally the maximal gradient at 385 K) shows a better agreement (not depicted). Such a value would coincide with a maximum of the second derivative of PAN over PVU, but further observations would be required to examine this more closely; also the second derivative is more difficult to compute due to numerical instability.

Figure 8 also shows the location of the primary thermal tropopause derived from CRISTA temperature data, which better encapsulates the area of the positive PAN VMR anomaly. The thermal tropopause is determined according to the World Meteo-

rological Organization (WMO) criterion (WMO, 1957). We first compute the lapse rate in between derived temperature values





and assign these lapse rates to the altitude of the mid-point of the interval. We then locate the altitude, where the lapse-rate falls below $2\,\mathrm{K\,km^{-1}}$, by linear interpolation, which gives a better result (also for model data) than simply assigning the tropopause altitude to the altitude of an available measurement point. Montgomery streamfunction isolines are also given in Fig. 8 and indicate the direction and speed of the geostrophic wind (the wind follows the isolines and is faster where they are closer

together). The wind approximately follows the thermal tropopause isoline and reaches its maximum at the thermal tropopause location except for the region between the ASM anticyclone and the shed eddy. This suggests that both the vertical and the horizontal transport barrier is related to the temperature structure of the anticyclone. The agreement is less good close to the cut-off point of the eddy and on the southern side of the anticyclone. The latter may be caused by an insufficient amount of data to properly determine the thermal tropopause. The region above India is very cloudy, preventing the retrieval of temperature

and trace gases in most of the troposphere, which obviously complicates the determination of the tropopause altitudes.

A better picture of the horizontal barriers is obtained by vertical cross-sections at fixed longitudes as shown in Fig. 5. The two cross-sections pass through the anticyclone and the separated eddy. At $90^\circ\,\mathrm{E}$, the thermal tropopause seems to provide a good transport barrier, or proxy thereof, for the PAN anomaly at altitudes above $14\,\mathrm{km}$. While the isentropes crossing the thermal tropopause would principally allow for the transport of PAN into the stratosphere, the strong jet inhibits this transport.

On the extratropical side beyond $40^\circ\,\mathrm{N}$, one can see elevated PAN VMRs for $\approx 3\,\mathrm{km}$ above the thermal tropopause, which indicates in-mixing of polluted tropospheric air into the extratropical lowermost stratosphere.

At least in the given situation, isentropes below $370\,\mathrm{K}$ do not cross the thermal tropopause on the southern side and remain at low PV values down to the equator. The equatorial jet also poses a barrier, but at a more southern position and not discernible by the PV criterion, which only works at higher latitudes. Measurements of PAN at $360\,\mathrm{K}$ show a stark contrast of clean air

south of the jet and polluted air north of it.

A different situation is given in the eddy, where elevated PAN VMRs can be found for $\approx 1\,\mathrm{km}$ above the thermal tropopause, however the altitudes and isentropes at which the PAN anomaly is found are still similar to the main anticyclone. This may indicate a horizontal intrusion of tropospheric air into the lowermost stratosphere across the thermal tropopause upon leaving the area of elevated pressure over India (see geopotential altitude in Fig. 2).

Another way of identifying the extent of confinement is by transferring the geo-located measurement data into tracer-tracer space (e.g. Hintsa et al., 1998). As noted, PAN is a mostly tropospheric gas without stratospheric sources, and its stratospheric lifetime is too short to accumulate there to larger VMR values. It is therefore possible to identify tropospheric air masses by large VMRs of PAN. Using another trace gas such as ozone that exhibits large VMRs only in the stratosphere, one may identify the association of an air parcel with either the troposphere or the stratosphere solely by the VMR of the two trace gases. Using

CRISTA-2 derived $O_3$ values, one derives ideally an "L"-like shape in the UTLS region (e.g. Hoor et al., 2002; Pan et al., 2007) such as shown in Fig. 9a. One line of the "L" is thereby formed by air parcels of tropospheric origin, while the other is formed by air parcels of stratospheric origin. Parcels not falling on one of the branches are then assumed to be generated by mixing between the "pure" branches. The stratospheric branch is well visible in Fig. 9 with high $O_3$ VMRs and correspondingly low PAN VMRS. The tropospheric branch is less pronounced, probably because the highest PAN VMRs can be found close to the

tropopause in contrast to $O_3$, where the highest VMRs can be found significantly above. Mixing between tropospheric and





stratospheric air introduces so-called mixing lines; that are air parcels that lie on lines between the two branches. Analysing the position of the measured air parcels in the tracer-tracer space shows no apparent mixing lines for parcels with an $O_3$ VMR of more than 1.0 ppmv. This indicates that no STE occurs for these air parcels.

Identifying thresholds from Fig. 9a and based on a priori knowledge of typical VMRs for both of these trace gases, we define parcels with PAN VMR of less than 80 pptv to be chemically of stratospheric nature and air parcels with $O_3$ VMR of less than 0.12 ppmv to be chemically of tropospheric nature and use these to separate the measured air parcels into four quadrants. The threshold for $O_3$ is quite high, which may be caused by pollution in the Asian monsoon. The parcels in the lower left quadrant with lower $O_3$ VMRS are found in the tropical troposphere and were advected from the Pacific region and are thus likely devoid of pollution. The presented results are not sensitive to small variations of the threshold.

Having thus identified tropospheric and stratospheric air masses, one may assign air parcels with both elevated PAN and $O_3$ VMRs to the mixing region of the UTLS. Figure 9b shows the number of such air parcels projected by trajectories to the 11th August of 2011. One can see that the number of such parcels (which effectively relates to the thickness of the extratropical mixing layer) is largest in the extratropical UTLS that is located roughly northward of the thermal tropopause on 380 K. The remaining anticyclone has very few such parcels, whereas the shed eddy exhibits a larger number as expected from the previous results.

It is apparent that the majority of mixing does not occur within the anticyclone, but within the shed eddy and in the transition region in between the two structures.

## 3.4 Source regions

The question remains, where and when the measured PAN found below the thermal tropopause originates and if it truly functions as a tracer for anticyclone air. To work as a tracer, it needs to originate from within the Asian monsoon region. Using the CLaMS model, we calculated backward trajectories for all parcels using ERA-Interim wind data. As criteria for air originating from the core of the anticyclone, we selected only air parcels with a PV value less than 3.7 PVU, a PAN VMR of greater than 150 pptv, and a potential temperature greater than 360 K. The latter restriction is necessary to exclude polluted air from the subtropics. Figure 10a shows the horizontal location of selected air parcels, which aligns well with the area of high PAN VMRs on 380 K shown in Fig. 8.

Following the trajectories backwards in time, we stopped the calculation as soon as the altitude of the parcel fell below 5 km — the results are however qualitatively similar for different thresholds such as 3 km. The majority of parcels can be traced back to the southern slopes of the Himalaya. There are several parcels stemming from the Pacific, which are likely mixed into the anticyclone on its southern side. This is consistent with a more comprehensive trajectory study by Bergman et al. (2013) and the CLaMS three-dimensional simulations using tracers of air mass origin by Vogel et al. (2015), who also found that a majority of air within the anticyclone stems from a very small region within India and China. In contrast, about half the air parcels subject to the same criteria except that we require a PAN VMR of below 100 pptv, stem from the Pacific region (not depicted), i.e. apparently a region where clean air originates.



The trajectories also allow an identification of the point in time, at which the air masses left the boundary layer to be drawn upwards into the anticyclone. Figure 11 shows the number of parcels that passed the 5 km threshold at the given day. Most air parcels left the boundary layer between the 20th of June and the 14th of July. Inspecting, for example, the peak on the 10th of July shows that the updraught captured in ERA-Interim reanalysis data seems to be caused by southerly winds blowing against the slope of the Himalaya. However, as convection is likely a major mechanism for transporting air into the UT, which is a mechanism that is not well captured in 6-hourly ECMWF reanalysis data, the question remains whether the air could origin from other sources over Asia apart from the ASM region.

However, the back-trajectories do not reveal significant sources outside the ASM region so that we can safely assume that the majority of measured PAN stems from polluted air from the Indian subcontinent.

## 4 Discussion and Conclusions

This study presented a highly resolved snapshot of PAN and $O_3$ VMRs in the ASM of the second week of August 1997. It represents the eldest dataset of globally derived PAN and has at the same time an unprecedented spatial resolution. It thus provides the most resolved snapshot of the ASM anticyclone currently available. The presented data and analysis provide further observations and evidence for the mechanisms that affect STE in the region of the ASM.

The observations show the ASM anticyclone and the shedding of a large eddy to the east. We could demonstrate that the air confined within the anticyclone and the eddy was highly polluted as indicated by increased PAN VMRs measured by CRISTA. The PAN anomaly was matched very well by PV values taken from ERA-Interim reanalysis data. We could show the relationship between PAN and PV as a function of PV and potential temperature and that there is a particular PV value for each isentrope where the decrease in PAN VMR is maximal. These PV values of strong PAN gradients increase with potential temperature up to the highest potential temperature found in the measured data. This demonstrates that PAN is enhanced within the anticyclone and the eddy compared to the stratospheric background and can therefore act as a marker of anticyclone air. The distribution of measured PAN indicates that the ASM anticyclone exhibits a strong confinement of contained air consistent with previous model studies and less well-resolved satellite observations (e.g. Park et al., 2007). We demonstrated that for the given situation the location of the thermal tropopause in an isentrope was a better marker for the extent of anticyclone air than the PV value derived from the maximal gradient (dPAN/dPV). This indicates that the thermal tropopause constitutes a strong transport barrier against vertical and horizontal transport into the lowermost stratosphere.

We further analysed the relationship between PAN and $O_3$ to identify the extent of mixing at the edges of the anticyclone and the eddy. As demonstrated, PAN can be used as a tracer for tropospheric air, while $O_3$ functions as a tracer for stratospheric air. Mapping all retrieved air parcels into tracer-tracer space allowed parcels to be identified that are chemically a mixture of both stratospheric and tropospheric air. Transferring these results from tracer-tracer space back into the geo-spatial space allowed the identification of the regions, where the strongest mixing was taking place. While the core of the anticyclone did not show many mixed parcels, the region in-between the eddy and the anticyclone showed many such parcels. The presented cross-sections show mixing of tropospheric air into the lowermost stratosphere above the eddy. This implies that the mechanism of





eddy shedding might be an exceptionally quick pathway for STE, where air previously confined of the ASM anticyclone may enter the UTLS mixing layer within days.

The sources of the PAN anomaly were identified by means of backward-trajectory simulations. The polluted air stems largely from the southern slopes of the Himalaya, being uplifted when winds blow to the mountains. In contrast, clean parcels could

largely be traced back towards the Pacific warm pool. The source regions for polluted and un-polluted parcels are consistently different. The majority of measured polluted air parcels within the ASM anticyclone stems from a rather small source region in the vicinity of the southern slopes of the Himalaya. Combined with STE by eddy shedding, the ASM provides a very rapid horizontal transport pathway of Asian pollution into the extratropical lowermost stratosphere.

The presented data serve well as an example for the capabilities of highly-resolving limb-sounding instruments for the

study of dynamic phenomena. Further insights could be gained by a better coverage in time (having the same number of measurements within 24 hours) and with a better vertical sampling in the UTLS, which would be feasible with a next-generation infrared limb-imager (e.g. Riese et al., 2005).

*Acknowledgements.* We sincerely thank A. Dudhia, Uni. Oxf., for providing the Reference Forward Model (RFM) used to calculate the optical path tables required by our forward model. The European Centre for Medium-Range Weather Forecasts (ECMWF) is acknowledged

for meteorological data support. This work was supported in part by the European Commission under grant number StratoClim-603557-FP7-ENV.2013.6.1-2.

The service charges for this open access publication have been covered by a Research Centre of the Helmholtz Association.

## Appendix A: Retrieval

This appendix describes the level 2 processing used to derive the trace gas data used in this paper. It is not based on previous

CRISTA-2 processing, but derived from the level 2 processor used for the air borne successor instrument CRISTA-NF described by Ungermann et al. (2013) with some necessary changes to accommodate to the different measurement mode of the satellite mission and differences in the available level 1 data (calibrated spectra). For completeness sake, we give here a full overview.

Deriving trace gases from measured radiances poses a so-called inverse problem. Knowing the radiative transfer equations and the state of the atmosphere, it is straightforward to compute emitted and absorbed infrared radiation with a spectral sim-

ulation code. It is however more difficult to discern the state of the atmosphere from measured radiances. This problem is an ill-posed problem, as no solution may exist due to measurement errors of the instrument, it may not be unique, and it may vary greatly with measurement noise. As such, it is typically not possible to identify the correct atmospheric state that generated the measured spectra.

Instead, one approximates the ill-posed exact problem by a well-posed approximate one that incorporates a priori knowledge

about the solution, which — for limb-sounders — usually consists of smoothness criteria to prevent unnatural oscillations in the resulting trace gas profiles. Also, one does not require an exact reproduction of the measured spectra, but allows for deviations in the order of the expected noise to compensate for the measurement errors. Mathematically, the problem is reformulated as



a minimisation problem. Let $\boldsymbol{F} : \mathbb{R}^n \mapsto \mathbb{R}^m$ be the forward model that maps a discretised atmospheric state $\boldsymbol{x} \in \mathbb{R}^n$ to the set of all evaluated radiance measurements of one profile $\boldsymbol{y} \in \mathbb{R}^m$. Then the solution $\boldsymbol{x}_{\mathrm{f}}$ is defined by the minimum of the cost function

$$J(\boldsymbol{x}) = (\boldsymbol{F}(\boldsymbol{x}) - \boldsymbol{y})^{\mathrm{T}} \mathbf{S}_{\epsilon}^{-1} (\boldsymbol{F}(\boldsymbol{x}) - \boldsymbol{y}) + (\boldsymbol{x} - \boldsymbol{x}_{\mathrm{a}})^{\mathrm{T}} \mathbf{S}_{\mathrm{a}}^{-1} (\boldsymbol{x} - \boldsymbol{x}_{\mathrm{a}}). \tag{A1}$$

The matrix $\mathbf{S}_{\epsilon} \in \mathbb{R}^{m \times m}$ approximates the true measurement error covariance matrix of the instrument, which here is constructed as a simple diagonal matrix with a relative error of 1%. The matrix $\mathbf{S}_{\mathrm{a}}^{-1} \in \mathbb{R}^{n \times n}$ together with the vector $\boldsymbol{x_a} \in \mathbb{R}^n$ defines the a priori knowledge of the atmosphere. The a priori values for pressure, temperature, and water vapour are taken from ECMWF ERA-Interim data, PAN assumes a zero-profile, while the remaining entities are taken from the Remedios et al. (2007) climatology. The matrix $\mathbf{S}_{\mathrm{a}}$ is assembled as the sum of a matrix with the variances of the atmospheric quantities on

the diagonal and a first-order Tikhonov regularisation matrix (Tikhonov and Arsenin, 1977), which in effect computes the L2-norm of the first derivative computed by finite differences, again scaled with the variance. To reduce the bias introduced by this regularisation we scale the zeroth-order matrix with a factor of 0.1. The correlation lengths employed for the regularisation are summarised in Tab. 1.

The simulation uses the microwindows of Tab. 2 and takes the trace gases $C_2H_6$, $CCl_4$, $ClONO_2$, $CO_2$, CFC-11, CFC-12,

HCFC-22, CFC-113, CFC-114, $H_2O$, $HNO_3$, $HNO_4$, $NH_3$, $N_2O_5$, $NO_2$, $O_3$, OCS, and PAN into account. Target quantities are temperature, $CCl_4$, CFC-11, HCFC-22, CFC-113, $ClONO_2$, $H_2O$, $O_3$, PAN, and an aerosol background. We assume that the pressure is well enough modelled by ECMWF in the altitude region relevant for this paper, such that the error is negligible (for error estimates, we assume an error of 1%).

The inversion itself employs the JURASSIC2 model as the forward model and the JUTIL inversion library for the minimi-

sation of the cost function. While the given problems are of much smaller size than tomographic ones usually treated with this code (e.g. Kaufmann et al., 2015), the employed algorithms are still efficient. JURASSIC2 and JUTIL use a combination of C++ and Python to combine optimal efficiency with a high adaptability. JUTIL minimises the cost function with a truncated conjugate-gradient trust-region method that leverages the inherent regularisation properties (Hanke, 1995) of the conjugate gradient method to improve the convergence speed. Retrieving one profile requires $\approx 60\,\mathrm{s}$ including the computation of diagnostic

information such as errors due to various sources and resolution for typically six iterations of the minimiser on one core.



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





| target | correlation length (km) |
|---|---|
| temperature | 2 |
| CCl$_4$ | 2 |
| ClONO$_2$ | 8 |
| CFC-11 | 2 |
| CFC-113 | 8 |
| HCFC-22 | 8 |
| H$_2$O | 2 |
| O$_3$ | 32 |
| PAN | 0.1 |

**Table 1.** A list of used correlation length for regularisation.

| IMW | range (cm$^{-1}$) | IMW | range (cm$^{-1}$) |
|---|---|---|---|
| 0 | 777.35 – 778.75 | 6 | 796.95 – 797.65 |
| 1 | 784.00 – 785.05 | 7 | 808.15 – 809.55 |
| 2 | 787.15 – 790.95 | 8 | 809.90 – 813.05 |
| 3 | 791.35 – 791.70 | 9 | 820.40 – 821.45 |
| 4 | 794.15 – 794.85 | 10 | 834.75 – 835.45 |
| 5 | 795.55 – 796.60 | 11 | 844.90 – 835.45 |

**Table 2.** A list of employed integrated microwindows (IMW) and their spectral range.



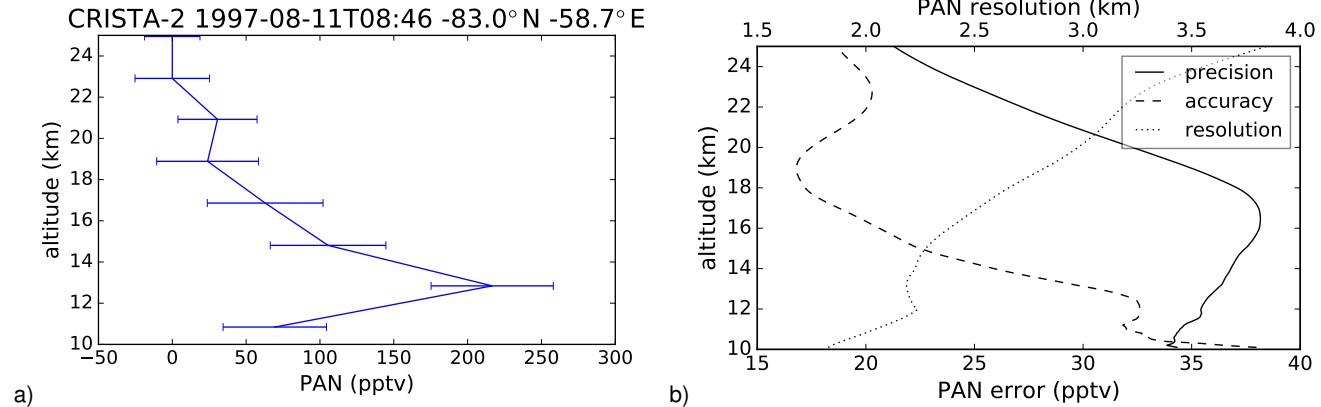

**Figure 1.** An exemplary retrieved PAN profile and diagnostic information. Panel (**a**) shows PAN on the retrieval grid of with the precision figures for this specific profile given as error bars. Panel (**b**) shows average diagnostic values for precision (measurement noise induced error), accuracy (other, likely systematic error sources, the leading one being elevation angle uncertainty, here), and resolution (full-width-half-max of averaging kernel matrix row).

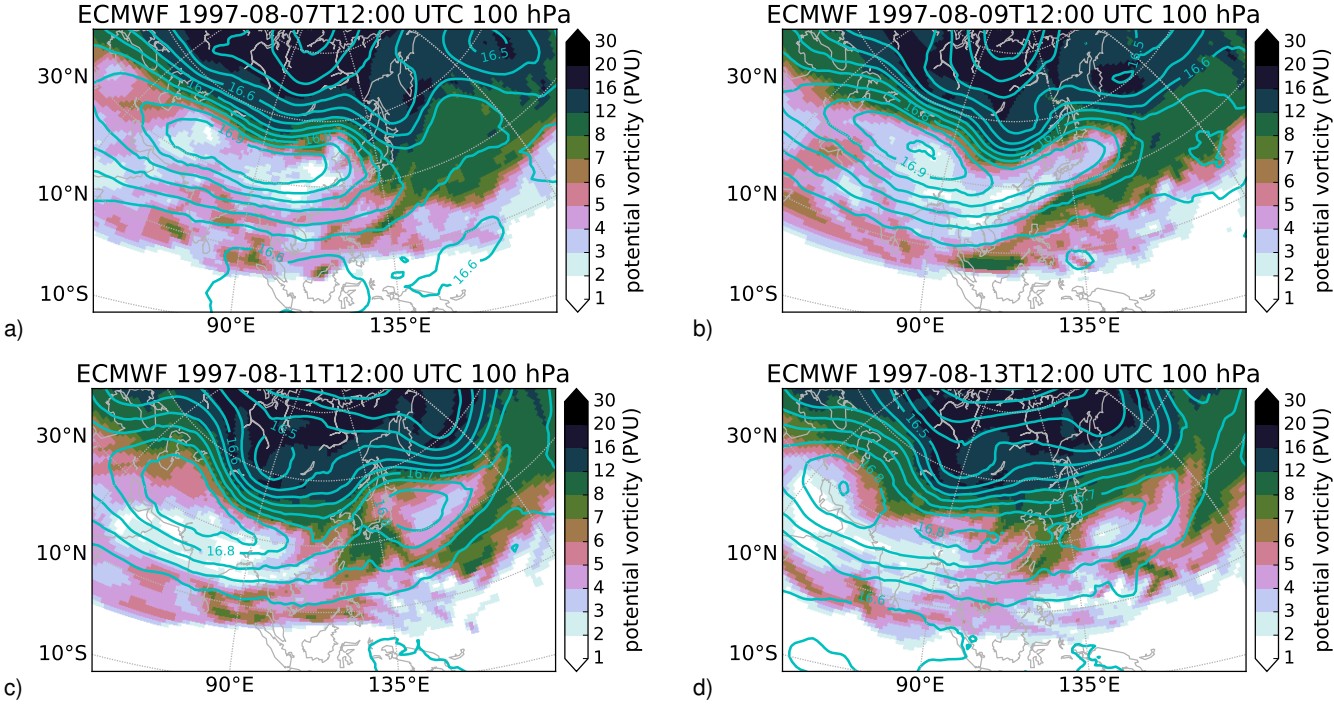

**Figure 2.** Potential vorticity (in PVU) derived from ECMWF ERA-Interim model data on 100 hPa. The contour lines show geopotential altitude (in km).





**Figure 3.** PAN derived from CRISTA-2 measurements on 380 K. Retrieved PAN VMRs are interpolated vertically to 380 K within each derived profile and located at the horizontal position of the closest tangent point. Shown are all values of the indicated day.



**Figure 4.** PAN derived from CRISTA-2 measurements and potential vorticity derived from ECMWF ERA-Interim model data on 380 K. The cyan contour lines indicate geopotential altitude. The PAN panels show PAN VMRs for all CRISTA-2 measurements taken from 9th to 13th August 1997. The horizontal location is determined by means of trajectory calculations to the position that the air parcels has at the time indicated by the title of the figure.



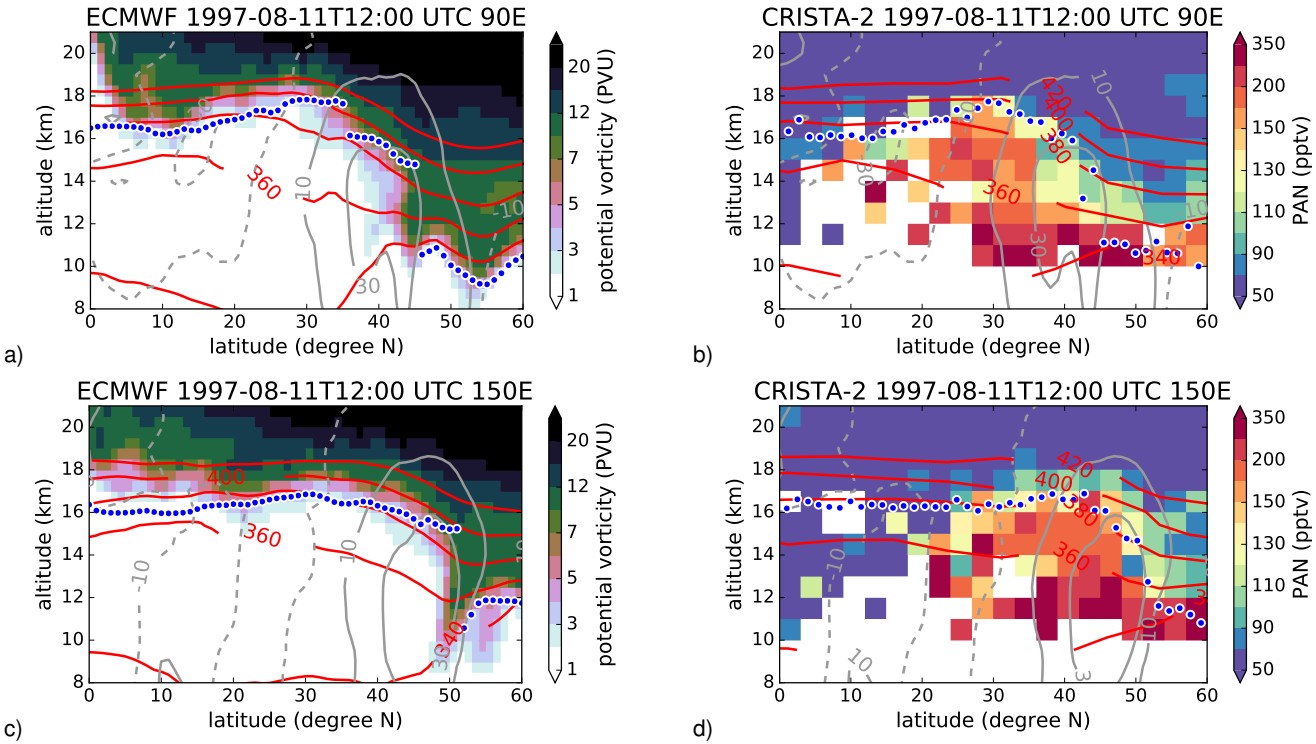

**Figure 5.** Vertical cross-sections along 90° E and 150° E. Panels **(a)** and **(c)** show potential vorticity with isentropes (red; K), winds (grey; ms$^{-1}$), and the primary thermal tropopause (blue dots) base on ERA-Interim data. Panels **(b)** and **(d)** show CRISTA PAN VMRs with CRISTA isentropes (red; K), ERA-Interim winds (grey; ms$^{-1}$), and CRISTA primary thermal tropopause (blue dots). The CRISTA data were averaged over ±5° longitude and missing pixels were horizontally interpolated to compute the CRISTA isentropes.





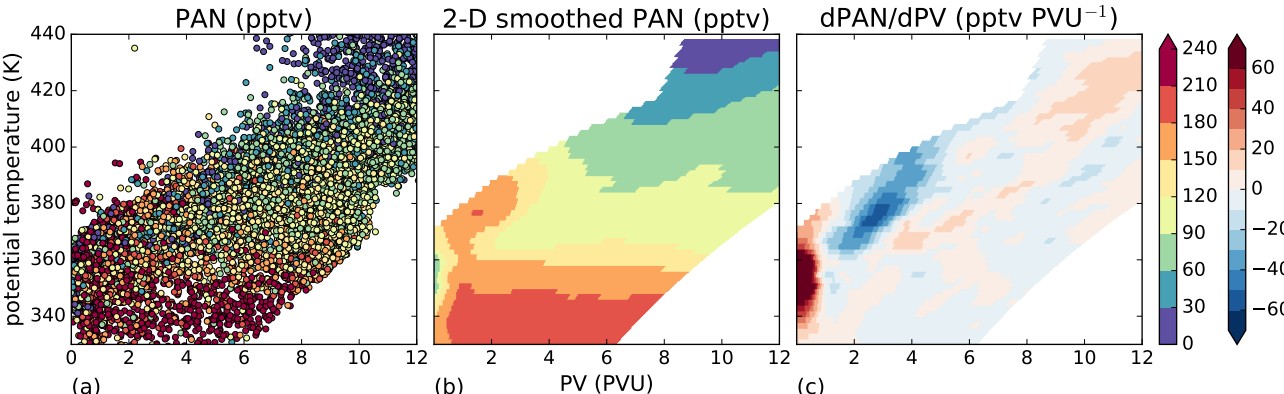

**Figure 6.** PAN VMRs against PV. Panel **(a)** shows measured potential temperature against PV with colour-coded PAN VMR for all air parcels between 20° E and 240° E, between 10° N and 60° N, and below 460 K; that is all pair parcels from the approximate location of the ASM and its surroundings. Panel **(b)** shows the fit of a smooth 2-D surface to the available data (the 5% most mis-fitting air parcels were removed in a first step; the surface is restricted to the area defined by measurements). Panel **(c)** shows the derivative of the smoothed PAN VMRs with respect to PV.



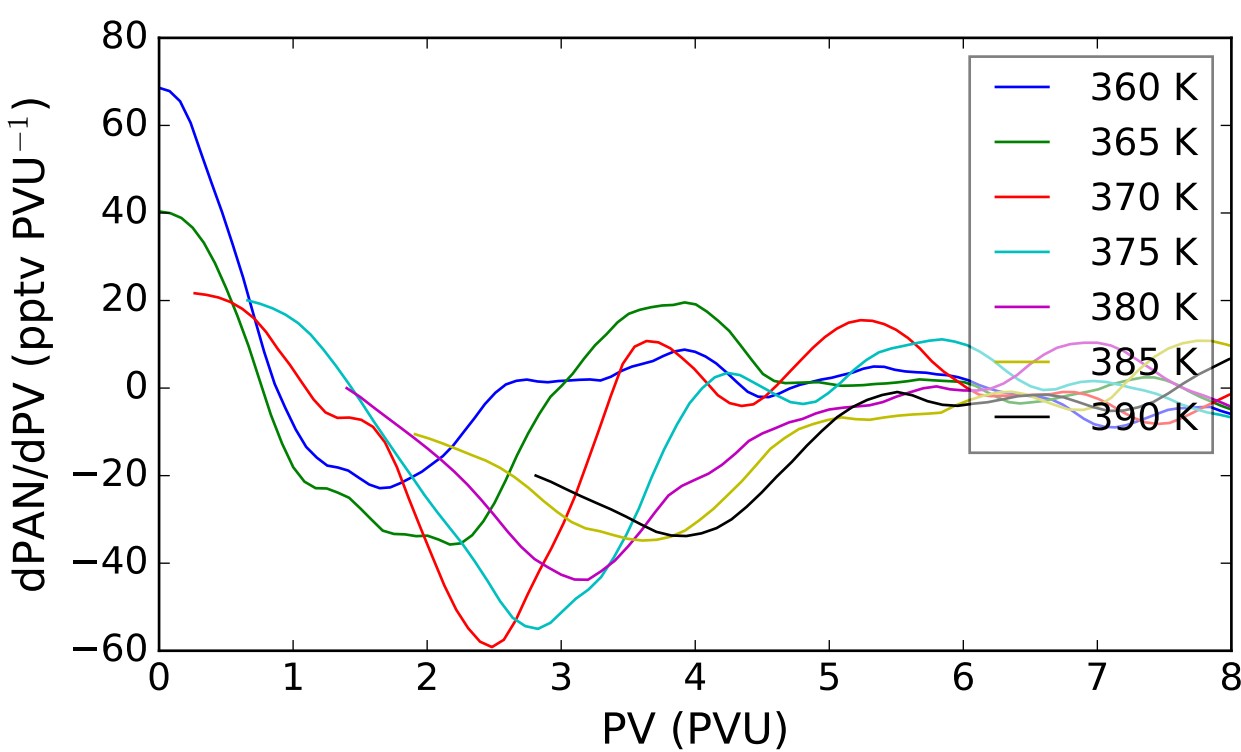

**Figure 7.** Derivative of smoothed PAN volume mixing ratios with respect to PV against PV, computed for all air parcels between 20° E and 240° E, between 10° N and 60° N, and below 460 K. This shows exemplary the derivative for selected isentropes of Fig. 6c.



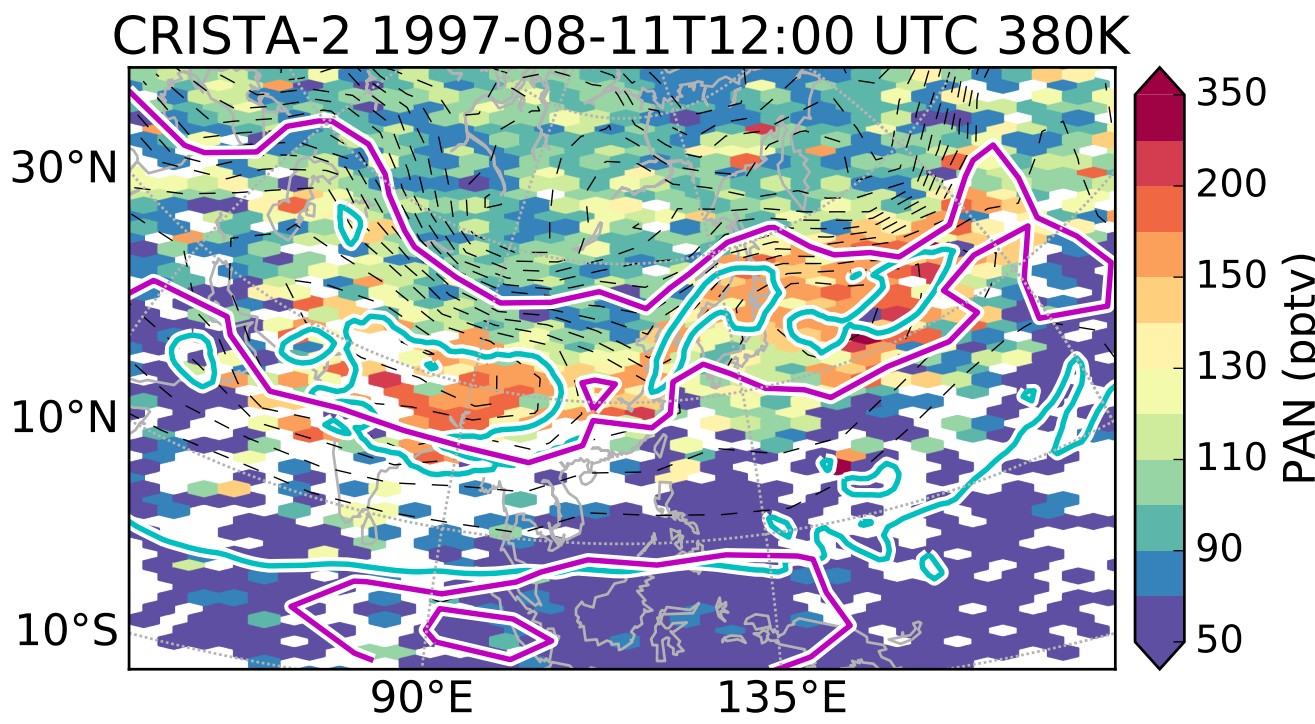

**Figure 8.** CRISTA PAN VMRs on 380 K. The PAN VMRs are as in Fig. 3d. Additionally shown are Montgomery streamfunction isolines (dashed black), Potential vorticity isoline (based on ERA-Interim) for 3.2 PVU (cyan), and the location of the CRISTA primary thermal tropopause (magenta).





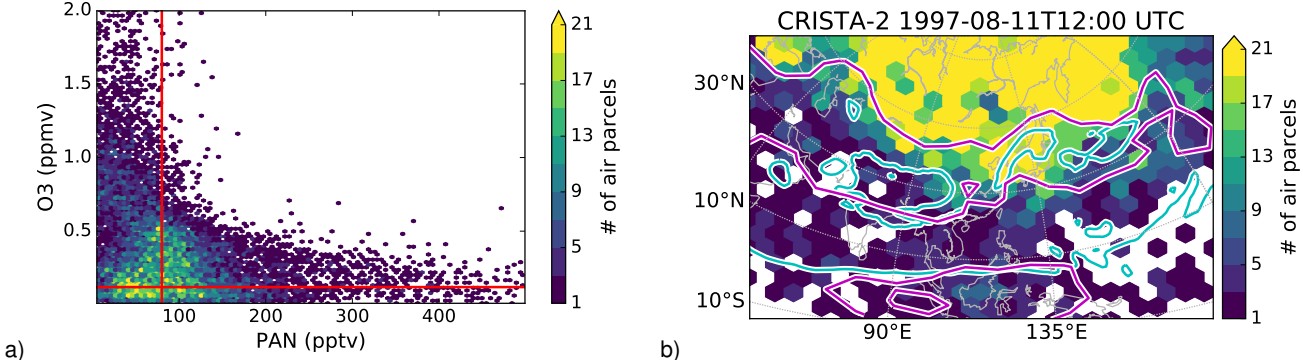

**Figure 9.** Panel **(a)** shows the relationship between PAN and $O_3$ for measured air parcels from $10°$ N to $80°$ N, from $20°$ E to $210°$ E, and with a distance of less than $5\,km$ from the thermal tropopause. Two red lines have been drawn at $80\,pptv$ PAN and at $0.12\,ppmv$ $O_3$ to roughly mark stratospheric (left) and tropospheric (bottom) branches. The high value of $0.12\,ppmv$ for $O_3$ may be caused by pollution in the Asian monsoon or be a systematic error of the CRISTA-2 retrieval; this is however not important for the qualitative analysis. Panel **(b)** shows the number of total air parcels that have more than $80\,pptv$ PAN and more than $0.12\,ppmv$ $O_3$. The cyan and magenta isolines are the same as in Fig. 8.

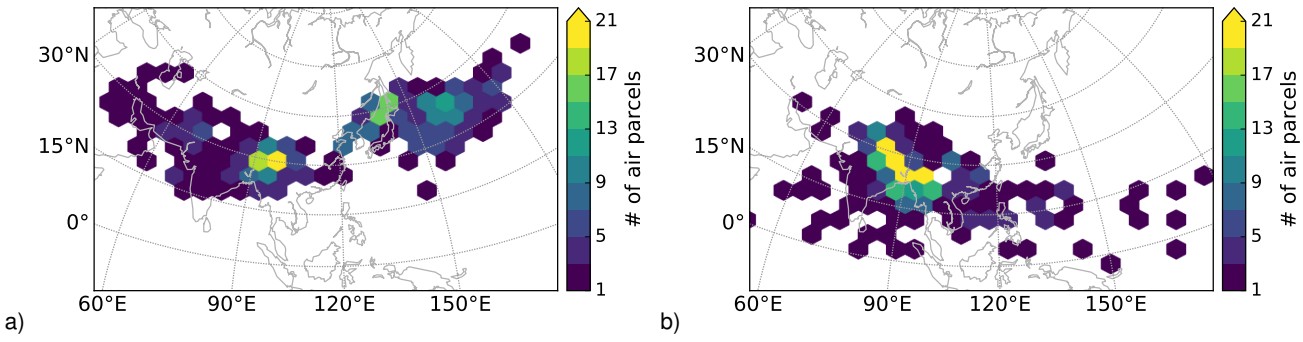

**Figure 10.** Panel **(a)** shows the number and horizontal location of all 454 parcels with PV less than 3.7, PAN VMR greater than $150\,pptv$, and potential temperature greater than $360\,K$ on 11th August 2011. Panel **(b)** shows the number and horizontal location of the subset of 378 air parcels, where the backward trajectories went below $5\,km$ altitude.




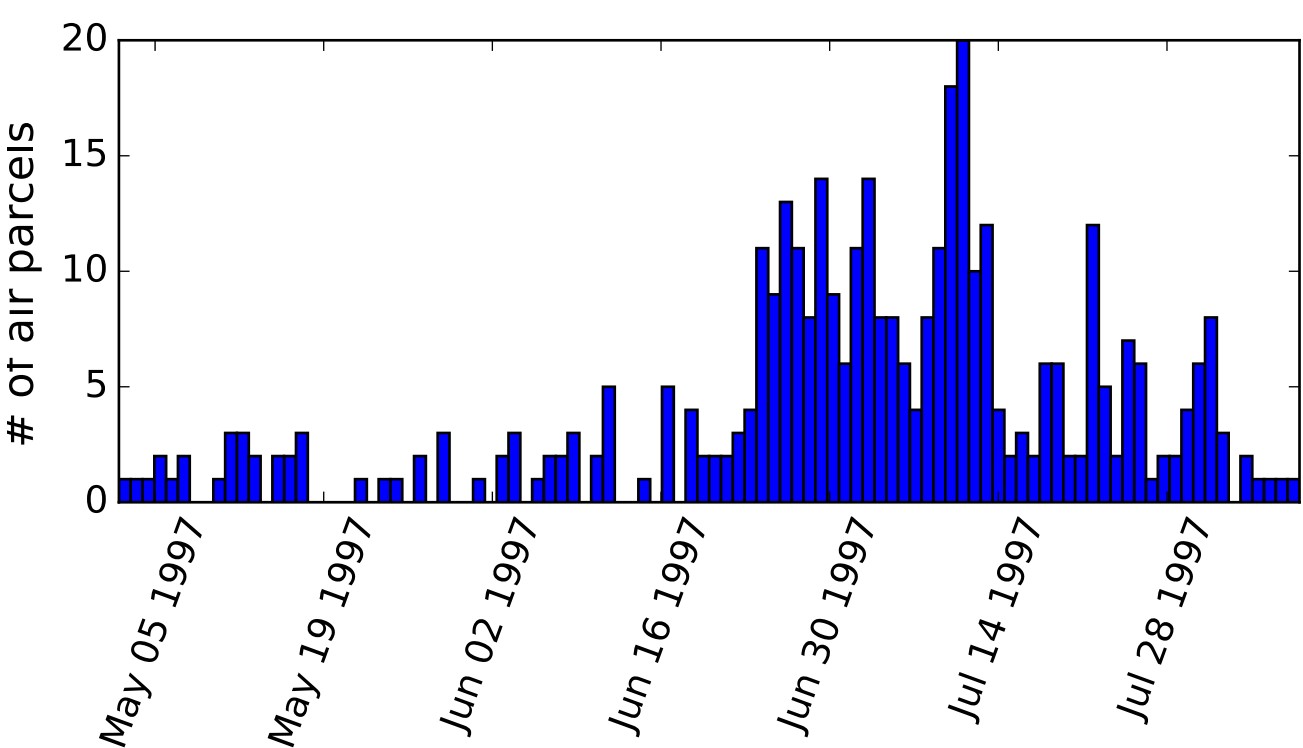

**Figure 11.** Histogram showing the point in time when the air parcels of Fig. 10 went below 5 km of altitude.