# Peer review of "Observations of PAN and its confinement in the Asian Summer Monsoon Anticyclone in high spatial resolution"

_Atmospheric Chemistry and Physics, 2016_

## Referee Comment (RC1) · L. Pan (Referee) · 1 Apr 2016

This discussion paper presents the data and analysis of peroxyacetyl nitrate (PAN), retrieved from the CRISTA-2 mission on the Space Shuttle during August 1997. This dataset is one-of-a-kind, providing a near instantaneous view of the UTLS chemical structure created by the dynamics of the Asian monsoon. The authors did an excellent job integrating the chemical and dynamical information, and complementing the direct observations with modeling tools. The work makes a significant contribution to the characterization of the ASM chemical impact. Some comments and suggestions are provided to the authors for improving the clarity of the analysis and the presentation.

Major points:

1. The most outstanding weakness of the discussion paper is the section on PAN-O3 tracer-tracer space analysis, captured by Figure 9. The idea of examining the tracer-tracer space is a very good one. The specific analysis and description presented need some critical thinking.

1) The vertical range of air masses represented in Figure 9: follow your statement, the air masses you selected for this analysis are within a 10 km layer following the tropopause ($\pm 5$ km around the tropopause). What is the horizontal range of the selection? Are they all from the region shown in Fig 9b? What is the altitude range represented in Fig 9b? i.e., the vertical distribution of the parcels in the O3>120 ppbv and PAN>80 ppbv quadrant? Does it justify the use of the 380K level tropopause isoline and PV contours?

2) Representation of the tropospheric branch: Based on Fig 5, sampling of the extratropical UT is very limited. That might be the main contributor to the ambiguity in identifying the tropospheric branch. It is important to look at the distribution from the specific dataset when selecting the critical values. Based on the tracer space distribution in Fig. 9a, there is no obvious reason for choosing 120 ppbv of ozone as a cut-off value for the tropospheric air. I am curious: what is the ozone distribution for parcels below the tropopause and equatorward of the subtropical jet core? Intuitively, the ozone cut-off seems to be more appropriate to be near 250-300 ppbv. The spatial resolution of the data here are significantly different from the aircraft in situ measurements. This factor needs to be considered when choosing the cut-off values.

3) Be careful and consider the difference between "the tropospheric (stratospheric) air" and "the air parcels of tropospheric (stratospheric) origin". They mean very different things. The former classifies the air mass and the branch. The latter says which layer the air is coming from but usually implies it has left that original layer. In this case (P7L32-32) I think you meant to say that the tropospheric branch is formed by tropospheric air, not by air parcels of tropospheric origin. Some clarification in wording is necessary.

4) Be aware of the difference between "where mixing occurred/was taking place" and "where mixed air was observed". You don't have enough information to prove the former so it would be more accurate to stay with the latter.

5) A similar idea but a different strategy is shown in Park et al., 2007 (Fig 9), which should be referenced here.

2. Additionally, I have minor concerns with the source region discussion.

1) You used "back trajectory falling to below 5 km" as the criterion to identify the source region. Please clarify how the terrain is treated in the trajectory calculation, since the Tibetan plateau is about 5 km in altitude. The criterion would have excluded the plateau as a source region. It would be better to use a terrain following criterion.

2) It would be useful to include a discussion of the surface wind field (southwesterly flow), which will give a perspective of the larger region that contributes to the entry point of the vertical lifting, not leaving the impression that only the southern slope of the Himalaya where the emission matters (a very small source region).

3. The paragraph starting P6L17 is somewhat troublesome. You intend to use the concept of tracer gradient to define the underlying dynamical barrier. You are struggling because the PAN-PV gradients do not seam to provide a clean result. Later you say it appears that the thermal tropopause works better than the PV contour selected from d(tracer)/d(PV) gradients. The fundamental problem is that PV itself is also a tracer and it goes through strong gradients at the barrier you are seeking, just like the chemical tracer. You will not find a strong gradient of one versus the other because they are co-varying. Please consider again the relevant information in Kunz et al. 2011a,b: 1) The tracer gradient should be calculated with respective to the latitude or equivelant latitude, and 2) at 380 K level, the separation of stratospheric and tropospheric air is better represented by 6pvu, not 2-4 pvu. If you look at Figure 4, a contour of 6 pvu may appear to be reasonably close to the contour of the thermal tropopause.

Specific wording suggestions and comments:

P3L14: "concentrate for practical matters" =>"focus on the information more relevant to this work"?

P3L18: "A simultaneous good signal-to-noise-ratio"?? => "a sufficient signal-to-noise-ratio"??

P3L25: "which could be observed" => "which was observed"

P4L2: "an aerosol underground" => "the background level of the aerosols" ??

P4L19: "the shown pressure surface" => "the 100 hPa pressure surface"

P4L31: "Shown is" => "Shown in Figure 3 is"

P5L1: "rather high in the anticyclone. . . trace gases" => "near the top of the anticyclone. The spatial extent of the anticyclone can be identified using the horizontal gradient in the trace gases at this level ."

P5L13: in this paragraph please be more specific about how many days of measurements you used to fill one day at 12Z using the trajectories. Is the filling only by forward or backward etc. Similarly, the paragraph in L25 could use some specific words like mid-point of every 3 days of measurements.

P6L12: "Figure 6c. . ." . This sentence is very confusing. It needs to be re-written.

P6L16 "tropical jet" => "tropical easterly jet" (check multiple occurrences)

P7L7: "is less good" => "is weaker"

P7L17: "in-mixing"=>"mixing"

P7L15: define "elevated"

P7L11: I think this paragraph is better to be following the Fig 5. The tracer cross sections show nicely that the tracer (PAN) is poleward bound by the westerly jet and

equator-ward bound by the easterly jet, although the easterly jet is weaker.

P7L21: the elevated PAN in the 150 °E cross-section looks very similar to the 90° E cross-section in its vertical extent.

P7L16: "majority of mixing does not occur within the anticyclone" – example of word "mixing occur" being used loosely. It is highly likely the mixing among the tropospheric air occurred within the anticyclone, but that kind of mixing is not what you designed to identify.

P8L26: Add "altitude" after "parcels fell below 5 km". As commented in point 3, there needs to be some clarification on how the terrain is treated in the back trajectory calculation.

P9L11: "It represents..." => The CRISTA-2 dataset represents the first high-resolution global coverage of PAN at the UTSL level."

P9L31: "where the strongest mixing was taking place" -see comment 1.4)

P10L6: "The majority ... stems from a rather small source region..." – see comments 2.2)

Figures: it will make it easier for the readers if the lat-lon great circles and the continents are draw with darker lines. Alternatively, change to using a different type of map projection. Current projection and the faint gray lines for the latitude circle made it very hard for the readers to compare the locations on vertical cross section (figure 5) with the maps, because the vertical cross sections extend to 60 °N latitude, the maps only show latitude labels up to 30°N.

References:

Kunz, A., P. Konopka, R. Müller, and L. L. Pan (2011a), Dynamical tropopause based on isentropic potential vorticity gradients, J. Geophys. Res., 116, D01110, doi:10.1029/2010JD014343.

[Figure]

Kunz, A., L. L. Pan, P. Konopka, D. E. Kinnison, and S. Tilmes (2011b), Chemical and dynamical discontinuity at the extratropical tropopause based on START08 and WACCM analyses, J. Geophys. Res., 116, D24302, doi:10.1029/2011JD016686.
* * *

---

## Referee Comment (RC2) · Anonymous Referee #2 · 18 Apr 2016

This paper presents spaceborne observations of peroxyacetyl nitrate (PAN) in the region around the Asian Summer Monsoon (ASM) with high spatial resolution. The major focus is to describe the relatively strong confinement of PAN in the ASM and the weaker confinement in an eastward travelling eddy. Further, it is tried to define criteria for the boundary of the ASM and to determine the source regions of the air masses in the ASM. Spaceborne observations of pollutants in the ASM region presented so far have mostly been averaged over longer time periods and had a coarser spatial resolution. Thus, the paper contains new observations that will be of interest to the ACP readership. However, in my opinion the data analysis contains some weaknesses and the wording should generally be improved. Thus, a careful revision is required,

before I can recommend publication in ACP.

**Major points:**

I have concerns about the benefits of Figs. 6 and 7 and the related discussion. On the one hand the authors want to point at different strengths of confinement in the ASM and in the eddy, but in Fig. 6 they obviously plot all PAN data versus PV and potential temperature. By means of Figs. 6c and 7 they try to derive criteria to visualize "which air masses are entrapped within" the ASM by inspection of dPAN/dPV. As shown in Figs. 4 and 5, both PV and PAN act as dynamical tracers, both having strong gradients at the boundaries of the ASM and of the eddy. Thus, I doubt if the derivative dPAN/dPV is a quantity, which is well suited to determine the boundary of the anticyclones. These doubts are confirmed in the discussion of Fig. 8, where the authors concede, that the curves of maximum gradient dPAN/dPV do not enclose the areas of enhanced PAN or of the anticyclones, but considerably smaller regions instead. Afterwards they resume the discussion of Fig. 5, giving "a better picture of the horizontal transport barriers."

I also have concerns about Fig. 9 and the related discussion. While the vertical red line in Fig. 9a indicating the boundary of the stratospheric branch seems to be appropriate, the threshold for the tropospheric branch at 0.12 ppmv ozone seems to be much too low for the actual data set. The data points in Fig. 9a suggest an at least two times higher value instead. Further, Fig. 9b shows that the major part of mixed air parcels is northward of the thermal tropopause at mid-latitudes. Where do these air parcels come from? From prior eddy shedding, from previous isentropic transport through the ASM boundary or is it tropospheric pollution originating at mid-latitudes?

**General remarks:**

The authors speak of "isentropes" throughout the manuscript. I think, when discussing horizontal distributions, they should rather speak of "isentropic levels" or "isentropic surfaces" instead.

2. Measurement and model data: The authors emphasise the unprecedented spatial resolution of their measurements as compared to other space borne observations. To substantiate this statement they should specify the along and across track sampling the PAN distribution is actually basing upon.

3.1 Synoptic situation: The authors might give some additional information on previous observations of eddy shedding from the ASM.

P5, L12-15: I do not completely understand how the authors manage to "... improve the measurement density by synoptically interpolating the measurements of multiple days to a single point of time ...". This procedure should be described more clearly.

P5, L25-27: I have also difficulties in understanding this sentence. Please describe the procedure more clearly.

P5, L29f: "The 380 K isentrope is well suited in the given meteorological situation to describe the confinement of the polluted air masses of the Asian monsoon." I do not understand the motivation for this sentence as introduction of Fig. 5, which is a latitude-height cross section.

P6, L12: The authors should add a description of the features visible in Fig. 6b (in case they want to maintain this Figure).

P6, L29: Does CO really have a longer lifetime than PAN in the UT?

P7, L11-16: This paragraph is somewhat contradictory. First the authors state that "... the thermal tropopause seems to provide a good transport barrier" and afterwards they note that "on the extra tropical side beyond 40°N, one can see elevated PAN VMRs for ≈3 km above the thermal tropopause."

P8, L24f: I do not quite understand, what exactly is depicted in Fig. 10. I assume the authors want to show the origin of the air parcels observed on 11 August by use of backward trajectories. They should describe the procedure more thoroughly to enhance clarity.

P9, L11-14: The information in this paragraph is somewhat redundant.

P10, L3-8: This paragraph contains redundant information on the origin of the polluted and unpolluted air masses. Please replace by a more concise phrasing.

**Specific comments:**

P1, L5: "within the ASM" is redundant and might be omitted.

P2, L2: "... has a major impact on trace gas composition". Please specify, in which regions.

P2, L5: "It can be found ..." instead of "It can be found in summer ...". It has been mentioned in the sentences before that the ASM prevails during summer.

P2, L30: Please add some more pioneering papers on PAN measurements.

P3, L4f: I think, Glatthor et al. (2007) covered the time period October to December 2003, only.

P3, L6: "... extends our knowledge on the historical evolution of PAN in the UTLS considerably". Considering a snapshot of 5 days, I think this conclusion is somewhat overdoing.

P4, L23: Is "signal-to-noise" ratio the appropriate term in this context?

P6, L12f: "increases linearly" instead of "increases smoothly and constantly", "between around 360 and 400 K" instead of "up to around 400 K".

P6, L18: "decrease of $\approx$-60 pptv PVU$^{-1}$ at $\approx$ 2.5 PVU".

P9, L17-18: "... the relationship between PAN and PV as a function of potential temperature ..." instead of "... the relationship between PAN and PV as a function of PV and potential temperature ..." ?

P11, eq. A1 and L5: I can not decipher the index of the measurement error covariance matrix.

**Wording:**

I am not a native English speaker, but I fear that there are various passages, where the wording is unclear or redundant. Some examples:

P1, L1: "This paper presents a set of observations by the CRISTA infrared limb sounder in low-earth orbit taken in August 1997 and analyses of trace-gases in the Asian Summer Monsoon (ASM) region." is kind of redundant and could be rephrased as follows: "This paper presents an analysis of trace gases in the Asian Summer Monsoon (ASM) region on basis of observations by the CRISTA infrared limb sounder taken in low-earth orbit in August 1997."

P1, L5ff: "Comparing the retrieved PAN VMRs with potential vorticity (PV) on isentropes reveals that the PAN VMRs exhibit the strongest decrease at each isentrope for an increasing value of PV, which may be used to identify the extent of the ASM" is difficult to understand. I suggest "Plotting the retrieved PAN VMRs against potential vorticity (PV) and potential temperature reveals that the PV value, at which the PAN VMRs exhibit the strongest decrease with respect to PV, increases with potential temperature. These PV values might be used to identify the extent of the ASM."

P1, L7ff: "... we also computed the location of the thermal tropopause ... and find that its location agrees well with the limits of the area of increased PAN VMRs both horizontally on isentropes and vertically within the anticyclone." I do not quite understand this sentence. Do the authors want to state that the thermal tropopause confines the area of enhanced PAN towards the stratosphere and towards mid-latitudes? Please clarify.

[Figure]

P5, L31f: "For both PV and PAN, the northern boundary (of what, enhanced PAN and low PV, I assume?) is formed by the jet-stream with strong winds of more than 30 ms$^{-1}$ following (followed by?) a sharp increase in PV (and low PAN amounts north of the jet?)."

P5, L32f: "The southern boundary (latitude) coincides with the thermal tropopause on the isentrope (altitude) close to the equatorial jet". I do not understand this sentence.

P6, L2: "Figure 6a does not show "a scatter plot of PAN VMRs against PV", but rather shows PAN VMRs plotted against PV and potential temperature instead. The same applies for the caption of Fig. 6.

P6, L20f: I do not quite understand, how "the southern transport barrier" (latitude) can coincide "with the thermal tropopause", which is more or less horizontal at these latitudes.

P6, L23f: "Comparing the PV value for the maximum gradient (dPAN/dPV) at 3.2 PVU with the given PAN VMRs ..." sounds strange. I suggest to rephrase as follows: "Comparing the PV isoline of the maximum gradient dPAN/dPV (3.2 PVU) with the given PAN VMRs ..."

P6, L26f: "The PAN VMRs may be not well-mixed enough within the anticyclone, leading to a displacement between the steepest decrease in VMR and the ASM boundary." instead of "The PAN VMRs may be not well-mixed enough within the anticyclone such that the steepest decrease in VMR coincides with the ASM boundary."

P7, L6f: "This suggests that both the vertical and the horizontal transport barrier is related to the temperature structure of the anticyclone." In which way? I do not quite understand this sentence.

P9, L13-14: "The presented data and analysis provide further observations and evidence ..." is redundant.

P9, L22: "... that air masses are strongly confined within the ASM anticyclone ..." instead of "... that the ASM anticyclone exhibits a strong confinement of contained air ..."

**Spelling**

P3, L28: "RApid" instead of "Rapid"?

P3, L30: "JUelich" instead of "Juelich"?

P4, L16: "The approximate location of the ASM is located". Please omit "located".

P4, L17: "... a clear separation between air masses of the main anticyclone and of the smaller eastward propagating anticyclonic eddy ..." instead of "... a clear separation of air masses between the main anticyclone and the smaller eastward propagating anticyclonic eddy ..."?

P4, L27f: "which is possible with the configuration" instead of "owing to the configuration"?

P5, L1: "... which typically is located in the upper part of the anticyclone ..." instead of "... which typically is located rather high in the anticyclone ..."

P5, L2: "... due to the expected strong horizontal gradients ..." instead of "... due to expected high horizontal gradient ..."

P5, L10: "Further note ... " instead of "Note further ..."

P6, L14: "and shows which air masses are entrapped within." instead of "and which air masses are entrapped within."

P8, L19: "... where and when originates ..." ?

P9, L17: "... matched very well to ..." instead of "... was matched very well by ..."

P10, L22: "For the sake of completeness ..." instead of "For completeness sake ..."

P10, L31f: "... deviations in ..." instead of "... deviations of ..."

Figure 1: "exemplarily" instead of "exemplary"?, "... with the precision given as error bars." instead of "... of with the precision figures for this specific profile given as error bars."

Figure 5: "based" instead of "base"

Figure 9: "the total number of air parcels" instead of "the number of total air parcels"?

Figure 10: "fell below" instead of "went below"?

Figure 11: "passed the altitude of 5 km" instead of "went below 5 km of altitude".

[Figure]

---

## Author Comment (AC1) · 30 May 2016

We thank the reviewers for their very helpful comments and insightful suggestions. Following these, we improved the treatment of the tracer-tracer-relationship analysis and separation of air parcels into the tropospheric and stratospheric branches and improved the paper overall. A point-by-point answer and a revised paper incorporating the comments is included in the attachement to this post.

Please also note the supplement to this comment:
http://www.atmos-chem-phys-discuss.net/acp-2016-34/acp-2016-34-AC1-supplement.pdf

---

## Author Response (AR1)

We thank the reviewers for their very helpful comments and insightful suggestions. Following these, we improved the treatment of the tracer-tracer-relationship analysis and separation of air parcels into the tropospheric and stratospheric branches and improved the paper overall. In the following we repeat the comments of the reviewers together with our reply and often also excerpts of the modified paper, where appropriate. We do not repeat minor technical suggestions, e.g. for wording, unless we did *not* apply them straightforwardly. Attached in the end is the revised paper with additions/deletions marked in red and blue colours.

**1 Reply to Referee #1**

**1.1 Major Comments**

1. The vertical range of air masses represented in Figure 9: follow your statement, the air masses you selected for this analysis are within a 10 km layer following the tropopause ($\pm 5$ km around the tropopause). What is the horizontal range of the selection? Are they all from the region shown in Fig 9b? What is the altitude range represented in Fig 9b? i.e., the vertical distribution of the parcels in the O3>120 ppbv and PAN>80 ppbv quadrant? Does it justify the use of the 380K level tropopause isoline and PV contours?

The horizontal range for the parcels of Fig. 9 and both panels is given in the caption and should be restricted to 10 to 80 degree N and 20 to 210 degree E. That some mixed parcels are found outside this domain in Fig. 9b is caused by the synoptic interpolation. However, due to a coding error, the lower boundary was actually set to -10 degree latitude, which has been corrected with updated figures (see also next item).

Within the region of the ASM, most parcels are located between 350 and 400 K (12 and 18 km) of potential temperature with a slight maximum between 360 and 380 K. Thus, the contour lines should have some relation to the depicted structures. However, another reason for plotting them is that they facilitate comparison of these structures with Fig. 8.

We added some clarifying sentences to the main text and modified the caption.

2. Representation of the tropospheric branch: Based on Fig 5, sampling of the extra-tropical UT is very limited. That might be the main contributor to the ambiguity in identifying the tropospheric branch. It is important to look at the distribution from the specific dataset when selecting the critical values. Based on the tracer space distribution in Fig. 9a, there is no obvious reason for choosing 120 ppbv of ozone as a cut-off value for the tropospheric air. I am curious: what is the ozone distribution for parcels below the tropopause and equatorward of the subtropical jet core?

Intuitively, the ozone cut-off seems to be more appropriate to be near 250-300 ppbv. The spatial resolution of the data here are significantly different from the aircraft in situ measurements. This factor needs to be considered when choosing the cut-off values.

We revisited the determination of the cut-off values according to your suggestions and use a more complicated but better matching approach based on linear regression.

We added an appropriate description for the determination of the thresholds and updated the figures in the paper:

*Analysing the position of the measured air parcels in the tracer-tracer space shows no apparent mixing lines for air parcels with an $O_3$ VMR of more than 1.0 ppmv. This indicates that no STE occurs for these air parcels. To identify the thresholds for separating the tropospheric/stratospheric branches from the "mixing region", we performed a linear regression. To analyse the stratospheric branch, we first selected all measured air parcels in the region of the ASM (from 10° N to 80° N and from 20° E to 210° E, located between the thermal tropopause and 5 km above, and with an $O_3$ VMR of less than 1 ppmv to exclude air parcels from the overworld). Using a linear regression, we identified a linear relationship for the middleworld between PAN and $O_3$ VMRs of: $PAN\,[mol\,mol^{-1}] = 1.11{\cdot}10^{-10}\,mol\,mol^{-1} - 0.7 \cdot 10^{-5} \cdot O_3\,[mol\,mol^{-1}]$. The residuals show a 1-sigma uncertainty of ≈34 pptv, which we added to the linear fit to determine the threshold as shown in Fig. 9a. A similar analysis of the tropospheric branch gave a relationship of $O_3\,[mol\,mol^{-1}] = 1.24{\cdot}10^{-7}\,mol\,mol^{-1} - 9.0{\cdot}10^{7}{\cdot}PAN[mol\,mol^{-1}]$ with a sigma of ≈0.06 ppmv.*

3. Be careful and consider the difference between "the tropospheric (stratospheric) air" and "the air parcels of tropospheric (stratospheric) origin". They mean very different things. The former classifies the air mass and the branch. The latter says which layer the air is coming from but usually implies it has left that original layer. In this case (P7L32-32) I think you meant to say that the tropospheric branch is formed by tropospheric air, not by air parcels of tropospheric origin. Some clarification in wording is necessary.

We agree that the wording is incorrect and changed the noted line. We also verified the correct use of stratospheric/tropospheric in the remainder of the document.

4. Be aware of the difference between "where mixing occurred/was taking place" and "where mixed air was observed". You don't have enough information to prove the former so it would be more accurate to stay with the latter.

We agree that the wording is highly imprecise and improved upon it along your suggestion at the noted place and also at other places in the document.

A similar idea but a different strategy is shown in Park et al., 2007 (Fig 9), which should be referenced here.

Park et al indeed uses a similar concept, with the major difference being that they do not try to distinguish the actual mixing from the overlap of the tropospheric and stratospheric branch. We added another reference to that paper in the introduction of the branches.

**1.2 Minor Comments**

1. You used "back trajectory falling to below 5 km" as the criterion to identify the source region. Please clarify how the terrain is treated in the trajectory calculation, since the Tibetan plateau is about 5 km in altitude. The criterion would have excluded the plateau as a source region. It would be better to use a terrain following criterion.

This is indeed unfortunate. We experimented with several different threshold to determine the sensitivity of the results to the cutoff altitude, but didn't notice this. To remove any potential influence of the Tibetan mountain range, we increased the threshold to 10 km. This increases the size of the most intense source region, but all results stay qualitatively the same.

2. It would be useful to include a discussion of the surface wind field (south-westerly flow), which will give a perspective of the larger region that contributes to the entry point of the vertical lifting, not leaving the impression that only the southern slope of the Himalaya where the emission matters (a very small source region).

We added a sentence to the discussion of the up-drafts on the southern slope:

*Most air parcels left the boundary layer between the 20th of June and the 14th of July. Inspecting, for example, the synoptic situation for the peak of transition events visible for the 10th of July shows that the updraught captured in ERA-Interim reanalysis data seems to be caused by south-westerly winds blowing against the slope of the Himalaya. This implies that, even though the entry point into the upper ASM is well confined, most air originates from the boundary layer located southwards.*

3. The paragraph starting P6L17 is somewhat troublesome. You intend to use the concept of tracer gradient to define the underlying dynamical barrier. You are struggling because the PAN-PV gradients do not seam to provide a clean result. Later you say it appears that the thermal tropopause works better than the PV contour selected from d(tracer)/d(PV) gradients. The fundamental problem is that PV itself is also a tracer and it goes through strong gradients at the barrier you are seeking, just like the chemical tracer.

You will not find a strong gradient of one versus the other because they are co-varying. Please consider again the relevant information in Kunz et al. 2011a,b: 1) The tracer gradient should be calculated with respective to the latitude or equivalent latitude, and 2) at 380 K level, the separation of stratospheric and tropospheric air is better represented by 6pvu, not 2-4 pvu. If you look at Figure 4, a contour of 6 pvu may appear to be reasonably close to the contour of the thermal tropopause.

The paragraph was indeed troublesome. Your notion that PV is effectively a dynamic tracer was a very helpful minder. In effect, one should be able to perform a similar analysis between PV and PAN as has been done now between PAN and O3. One expects some kind of L-shape between the two. However, the strong dependence of PV on potential temperature requires basically a separate treatment for each theta level, even when using modified PV. While the given analysis is not perfect, it performs this job quite well. The difficult point is now to properly define the threshold between the "branches", as the previously used maximum gradient does not make sense. Fitting a higher polynomial through the branches should, ideally, give the highest curvature on the transition from one branch to the other. And The first derivative should change from one constant value to another one. As such the noted criteria of maximum in second derivative (curvature) should be a better one than the one chosen in the first draft. As we could not compute the second derivative satisfactorily due to numerical reasons, we choose now a threshold in between the "linear" regime outside the ASM and the steepest gradient. Under certain assumptions, this should align with the maximum of the second derivative. We also played around with actually fitting the branches with a linear regression on each level, but the amount of measurements in the tropospheric branch on each single level is too small for a sensible fit.

As can be seen in the updated graphics, this value functions quite well. A value of $6\,\mathrm{PVU}$, which according to (Kunz et al., 2011) might be sensible at this potential temperature, covers a much too large area, especially on the southern boundary of the ASM.

Due to the change of the threshold value, much of the section has been (re)-written. Please refer to the revised paper, where the changes are highlight by colour codes.

**1.3 Specific Comments**

1. P7L11: I think this paragraph is better to be following the Fig 5. The tracer cross sections show nicely that the tracer (PAN) is poleward bound by the westerly jet and equator-ward bound by the easterly jet, although the easterly jet is weaker.

We moved this and the following paragraphs to the beginning of the section to have the discussion of the vertical cross-sections at one place:

*Figure 5 shows two vertical cross-sections through the potential vorticity*

*field, one through the main anticyclone (Fig. 5a) and one through the eddy (Fig. 5c). For both PV and PAN anomalies, the northern boundary is formed by the jet-stream with strong winds of more than $30\,ms^{-1}$ followed by a sharp increase in PV and lower PAN VMRs north of the subtropical westerly jet.*

*In Fig. 5b at 90° E, the thermal tropopause seems to provide a good transport barrier, or proxy thereof, for the positive PAN anomaly at tropical latitudes. While the isentropic levels crossing the thermal tropopause would principally allow for the transport of PAN into the stratosphere, the westerly and easterly jets inhibit this transport. Still, on the extratropical side northward of 40° N, PAN VMRs are found to be elevated by $\approx$20 to 40 pptv for $\approx$3 km above the thermal tropopause indicating transport of polluted tropospheric air into the extratropical lowermost stratosphere, which may have entered this extra-tropical transition layer (exTL) during breaking Rossby waves or an eddy-shedding event.*

*In the given situation, isentropic surfaces below 370 K do not cross the thermal tropopause on the southern side of the ASM and remain at low PV values down to the equator. The tropical easterly jet also poses a horizontal transport barrier, but at a more southern position and not discernible by the PV criterion, which only works at higher latitudes. Measurements of PAN at 360 K show a strong contrast of clean air south of the tropical easterly jet and polluted air north of it within the ASM.*

*A different situation is given in the eddy in Fig. 5d, where elevated PAN VMRs can be found for $\approx$1 km above the tropical thermal tropopause; however the altitudes and potential temperatures at which the PAN anomaly is found are still similar to the main anticyclone. This may indicate horizontal transport of tropospheric air into the lowermost stratosphere across the thermal tropopause upon leaving the area of elevated pressure over India (see geopotential altitude in Fig. 2). Alternatively, the descending thermal tropopause may become more permeable and start forming a transition layer like in the extra-tropics.*

2.     P7L21: the elevated PAN in the 150°E cross-section looks very similar to the 90°E cross-section in its vertical extent.

Indeed, the major difference is in the thermal structure, where the thermal tropopause is located lower and where also the isentropes are shifted.

3.     Figures: it will make it easier for the readers if the lat-lon great circles and the continents are draw with darker lines. Alternatively, change to using a different type of map projection. Current projection and the faint gray lines for the latitude circle made it very hard for the readers to compare the locations on vertical cross section (figure 5) with the maps, because

the vertical cross sections extend to 60°N latitude, the maps only show latitude labels up to 30°N.

We reworked the figures. To ensure better visibility of the meridians and parallels, we employed darker shades of grey. We also added country-borders to give better association of airmasses with known landmarks and used also darker shade of grey for theses as well as for the coastlines. We further ensured that the meridians of 90°E and 150°E are labelled and drawn in the figures. We hope that these changes also allow the reader to more readily find the position of the cross-sections in the horizontal plots and to identify the parallel for 50°N even though it is not labelled.

**2 Reply to Referee #2**

**2.1 Major points**

1. I have concerns about the benefits of Figs. 6 and 7 and the related discussion. On the one hand the authors want to point at different strengths of confinement in the ASM and in the eddy, but in Fig. 6 they obviously plot all PAN data versus PV and potential temperature. By means of Figs. 6c and 7 they try to derive criteria to visualise "which air masses are entrapped within" the ASM by inspection of dPAN/dPV. As shown in Figs. 4 and 5, both PV and PAN act as dynamical tracers, both having strong gradients at the boundaries of the ASM and of the eddy. Thus, I doubt if the derivative dPAN/dPV is a quantity, which is well suited to determine the boundary of the anticyclones. These doubts are confirmed in the discussion of Fig. 8, where the authors concede, that the curves of maximum gradient dPAN/dPV do not enclose the areas of enhanced PAN or of the anticyclones, but considerably smaller regions instead. Afterwards they resume the discussion of Fig. 5, giving "a better picture of the horizontal transport barriers."

We reorganised the discussion slightly and updated the section discussing Figs. 6 and 7. Thinking of PV as dynamic tracer allows to interpret the graphics along similar ways of the tracer-tracer space and justifies a different threshold value to be chosen from the given data, which fits much better to the extent of the ASM.

Please see also our reply to the third Minor Comment of Reviewer #1.

2. I also have concerns about Fig. 9 and the related discussion. While the vertical red line in Fig. 9a indicating the boundary of the stratospheric branch seems to be appropriate, the threshold for the tropospheric branch at 0.12 ppmv ozone seems to be much too low for the actual data set. The data points in Fig. 9a suggest an at least two times higher value instead. Further, Fig. 9b shows that the major part of mixed air parcels is northward of the thermal tropopause at mid-latitudes. Where do these air parcels

come from? From prior eddy shedding, from previous isentropic transport through the ASM boundary or is it tropospheric pollution originating at mid-latitudes?

We revamped the analysis of the chemical assignment to tropospheric and stratospheric branch from a rather vague qualitatively analysis to a more robust quantitative one. Please note that we previously experimented with different thresholds and always got similar results for the location of the mixed parcels, so we chose in the end sensible values from an a priori expectation point of view.

We now derive linear relationships between PAN and O3 for the uppermost troposphere and the middleworld and derive the threshold by using these relationships in combination with the 1-sigma value derived from the fit. We also reduced the shown range of ozone values, enabling a better view on the relevant region. The newly derived relationships give obviously visually much more pleasing results. The location of identified "mixed" air parcels stays the same, even though the number of such categorised parcels is greatly diminished.

The "mixed" air parcels northwards of the subtropical jet-stream are found in nearly the whole northern hemisphere, not just over Asia. We thus assume that these form the extra-tropical mixing layer, which usually has a vertical extent of several kilometres and is thus much more visible from space than STE in the tropics. We extended the discussion in the paper:

> Having thus identified chemically tropospheric and stratospheric air masses, one may associate air parcels with both elevated PAN and $O_3$ VMRs with the mixing layer of the UTLS. Figure 9b shows the number of the mixed air parcels projected by trajectories to 12:00 UTC on the 11th August of 2011. The number of such parcels (which effectively relates to the thickness of the exTL) is largest in the extratropical UTLS that is located roughly northward of the westerly jet-stream on 380 K. Most of these air parcels are located between 10 and 14 km. The anticyclone has very few such air parcels, whereas the shed eddy and especially the small remaining connection in between the anticyclone and the eddy exhibit a larger number as expected from the previous results. This is further evidence that the anti-cyclone confines its air masses quite well and that mixing occurs during the eddy-shedding process or within the newly shed eddy.

**2.2   General remarks**

1. The authors speak of "isentropes" throughout the manuscript. I think, when discussing horizontal distributions, they should rather speak of "isentropic levels" or "isentropic surfaces" instead.

We adjusted the paper accordingly.

2. 2. Measurement and model data: The authors emphasise the unprecedented spatial resolution of their measurements as compared to other space borne observations. To substantiate this statement they should specify the along and across track sampling the PAN distribution is actually basing upon.

We added the sentence

*The typical along-track sampling for the three measurement tracks and the discussed measurements is ≈250 km while the typical across-track sampling is ≈600 km.*

3. 3.1 Synoptic situation: The authors might give some additional information on previous observations of eddy shedding from the ASM.

We added the following:

*Such smaller anticyclones typically break off several times from the main anticyclone during summer, even though westward propagating ones are more common (e.g. Dethof et al., 1999; Hsu and Plumb, 2000; Popovic and Plumb, 2001; Garny and Randel, 2013; Vogel et al., 2014).*

4. P5, L12-15: I do not completely understand how the authors manage to "... improve the measurement density by synoptically interpolating the measurements of multiple days to a single point of time ...". This procedure should be described more clearly.

We elaborated as

*For all air parcels measured by the CRISTA-2 instrument from the 9th to 13th August 1997, forward and backward trajectories were computed using the CLaMS trajectory model from the time of measurement to 12:00 UTC of each day. Using these trajectories, one can project all measured air parcels to the noon of any given day and thereby it is possible to obtain a coherent and more complete picture of the synoptic situation. Fig. 4 shows PV and synoptically calculated PAN data on 380 K potential temperature combined from all measurements.*

5. P5, L25-27: I have also difficulties in understanding this sentence. Please describe the procedure more clearly.

We elaborated as

*Using trajectories for the synoptic interpolation introduces an additional uncertainty to the data due to the uncertainties of available wind data. Even though this uncertainty is small for the analysed altitude range and*

*time-scale of about one week, we minimise its impact by choosing the mid-point of the measurement period, 12:00 UTC on the 11th August 1997, for further analysis to reduce the time over which trajectories need to be computed to less than 36 hours on average.*

6. P5, L29f: "The 380 K isentrope is well suited in the given meteorological situation to describe the confinement of the polluted air masses of the Asian monsoon." I do not understand the motivation for this sentence as introduction of Fig. 5, which is a latitude-height cross section.

Indeed. We moved the sentence to a later, more fitting, place.

7. P6, L12: The authors should add a description of the features visible in Fig. 6b (in case they want to maintain this Figure).

We added

*One can discern the general decrease of PAN VMRs with potential temperature. Horizontally, two features are notable. For low PV values at 350 K a negative PAN anomaly can be seen, which is caused by the inclusion of clean tropical air southwards of the easterly jet. More important is the decrease of PAN VMRs from about 370 K to 390 K, which is indicative of the uplift and confinement of PAN-rich air masses within the ASM anticyclone.*

8. P6, L29: Does CO really have a longer lifetime than PAN in the UT?

Close to the tropopause, the lifetime may indeed be comparable but in the boundary layer, at higher temperatures, the statement is true. However, the statement got removed during the reorganisation of the pot. temp./PV/PAN relationship discussion.

9. P7, L11-16: This paragraph is somewhat contradictory. First the authors state that "... the thermal tropopause seems to provide a good transport barrier" and afterwards they note that "on the extra tropical side beyond 40°N, one can see elevated PAN VMRs for $\approx$3 km above the thermal tropopause."

We reformulated more exactly as

*In Fig. 5b at 90° E, the thermal tropopause seems to provide a good transport barrier, or proxy thereof, for the positive PAN anomaly at tropical latitudes.*

10. P8, L24f: I do not quite understand, what exactly is depicted in Fig. 10. I assume the authors want to show the origin of the air parcels observed on 11 August by use of backward trajectories. They should describe the procedure more thoroughly to enhance clarity.

The intent of the figure is to determine the pathway/source region, from where the polluted air entered the upper ASM circulation - as far as this is feasible using trajectories.

We elaborated the description including a direct reference to the panel b:

> *Using the CLaMS model, we calculated backward trajectories for all parcels using ERA-Interim wind data. As criteria for air originating from the core of the anticyclone, we selected only air parcels with a PV value less than 3.7 PVU, a PAN VMR of greater than 150 pptv, and a potential temperature greater than 360 K. The latter restriction is necessary to exclude polluted air from the subtropics. Figure 10a shows the horizontal location of selected air parcels, which aligns well with the area of high PAN VMRs on 380 K shown in Fig. 8.*
>
> *Following the trajectories of the selected air parcels backwards in time, we stopped the calculation as soon as the altitude of the parcel fell below 10 km altitude — the results are however qualitatively similar for different thresholds such as 5 km altitude. Almost all air parcels fell below the threshold altitude in the preceding two month. Figure 10b shows the locations, where the back-tracked air parcels crossed the 10 km threshold. The majority of parcels can be thus traced back to the southern slopes of the Himalaya. Obviously, this defines only the entry point into the upper circulation of the ASM, not necessarily the point of origin on ground level. There are some parcels stemming from the Pacific, which are likely mixed into the anticyclone on its southern side.*

11. P9, L11-14: The information in this paragraph is somewhat redundant.

We streamlined the paragraph:

> *This study presented a highly resolved snapshot of PAN and $O_3$ VMRs in the ASM of the second week of August 1997. The CRISTA-2 dataset represents the first high-resolution global coverage of PAN at the UTLS level and perhaps the currently best available snapshot of the ASM.*

12. P10, L3-8: This paragraph contains redundant information on the origin of the polluted and unpolluted air masses. Please replace by a more concise phrasing.

We reduced the paragraph to:

> *The sources of the PAN anomaly were identified by means of backward-trajectory simulations. Most of the polluted air enters the anticyclone above the southern slopes of the Himalaya, being uplifted when south-westerly winds blow against the mountains. In contrast, clean air parcels could largely be traced back towards the Pacific warm pool.*

**2.3 Specific comments**

P11, eq. A1 and L5: I can not decipher the index of the measurement error covariance matrix.

The index is an epsilon "$\epsilon$".

**2.4 Wording**

1. P1, L7ff: "... we also computed the location of the thermal tropopause ... and find that its location agrees well with the limits of the area of increased PAN VMRs both hori- zontally on isentropes and vertically within the anticyclone." I do not quite understand this sentence. Do the authors want to state that the thermal tropopause confines the area of enhanced PAN towards the stratosphere and towards mid-latitudes? Please clarify.

We clarified as

*[...], we also computed the location of the thermal tropopause according to the WMO criterion and find that it confines the PAN anomaly vertically within the main ASM anticyclone.*

This implies also a confinement towards the southern edge on horizontal levels, where the thermal tropopause is continuous. Towards the subtropics, the situation is more complicated due to the possibility of multiple tropopauses and the break in tropopause altitude at the subtropical jetstream.

2. P5, L32f: "The southern boundary (latitude) coincides with the thermal tropopause on the isentrope (altitude) close to the equatorial jet". I do not understand this sentence.

We first clarified as

*The southern boundary on 380 K and above coincides well with the location of the thermal tropopause on the same isentrope close to the tropical easterly jet suggesting a relationship between the two.*

but then got rid of the (redundant) statement during the reorganisation of this section.

3. P6, L20f: I do not quite understand, how "the southern transport barrier" (latitude) can coincide "with the thermal tropopause", which is more or less horizontal at these latitudes.

We refer to the uppermost part of the ASM anticyclone above 370 K, where the isentropic surfaces and the surface defined by the thermal tropopause cross on the southern edge of the ASM.

However, we removed the misleading statement during the reorganisation of this section due to other comments.

4.    P7, L6f: "This suggests that both the vertical and the horizontal transport barrier is related to the temperature structure of the anticyclone." In which way? I do not quite understand this sentence.

We removed the confusing sentence.

**2.5  Spelling**

1.    P3, L28: "RApid" instead of "Rapid"?

P3, L30: "JUelich" instead of "Juelich"?

The letters used in the acronym are highlighted in upper case.

**References**

[revised manuscript text omitted]